# KICK BAD GUYS OUT!
# ZERO-KNOWLEDGE-PROOF-BASED ANOMALY DETECTION IN FEDERATED LEARNING

## ABSTRACT

Federated learning (FL) systems are vulnerable to malicious clients that submit poisoned local models to achieve their adversarial goals, such as preventing the convergence of the global model or inducing the global model to misclassify some data. Many existing defense mechanisms are impractical in real-world FL systems, as they require prior knowledge of the number of malicious clients or rely on re-weighting or modifying submissions. This is because adversaries typically do not announce their intentions before attacking, and re-weighting might change aggregation results even in the absence of attacks. To address these challenges in real FL systems, this paper introduces a cutting-edge anomaly detection approach with the following features: *i*) Detecting the occurrence of attacks and performing defense operations only when attacks happen; *ii*) Upon the occurrence of an attack, further detecting the malicious client models and eliminating them without harming the benign ones; *iii*) Ensuring honest execution of defense mechanisms at the server by leveraging a zero-knowledge proof mechanism. We validate the superior performance of the proposed approach with extensive experiments.

## 1 INTRODUCTION

Federated Learning (FL) (McMahan et al., 2017a) enables clients to collaboratively train machine learning models without sharing their local data with other parties, maintaining the privacy and security of their local data. Due to its privacy-preserving nature, FL has attracted considerable attention across various domains and has been utilized in numerous areas (Hard et al., 2018; Chen et al., 2019; Ramaswamy et al., 2019; Leroy et al., 2019; Byrd & Polychroniadou, 2020; Chowdhury et al., 2022). However, even though FL does not require sharing raw data with others, its decentralized and collaborative nature inadvertently introduces privacy and security vulnerabilities (Cao & Gong, 2022; Bhagoji et al., 2019; Lam et al., 2021; Jin et al., 2021; Tomsett et al., 2019; Chen et al., 2017; Tolpegin et al., 2020; Kariyappa et al., 2022; Zhang et al., 2022c). Malicious clients in FL systems can harm training by submitting spurious models to disrupt the global model from converging (Fang et al., 2020; Chen et al., 2017), or planting backdoors to induce the global model to perform wrongly for certain samples (Bagdasaryan et al., 2020b;a; Wang et al., 2020).

Existing literature on robust learning and mitigation of adversarial behaviors includes Blanchard et al. (2017); Yang et al. (2019); Fung et al. (2020); Pillutla et al. (2022); He et al. (2022); Cao et al. (2022); Karimireddy et al. (2020); Sun et al. (2019); Fu et al. (2019); Ozdayi et al. (2021); Sun et al. (2021), etc. These approaches exhibit shortcomings, making them less suitable for real FL systems. Some of these strategies require prior knowledge about the number of malicious clients within the FL system (Blanchard et al., 2017), even though in practice an adversary would not notify the system before attacking. Also, some of these methods mitigate impacts of potential malicious client submissions by re-weighting the local models (Fung et al., 2020), retaining only several local models that are most likely to be benign while removing others (Blanchard et al., 2017), or modifying the aggregation function (Pillutla et al., 2022). These methods have the potential to unintentionally alter the aggregation results in the absence of deliberate attacks, considering attacks happen infrequently in real-world scenarios. While the defense mechanisms can mitigate the impact of potential attacks, they can inadvertently compromise the result quality when applied to benign cases.

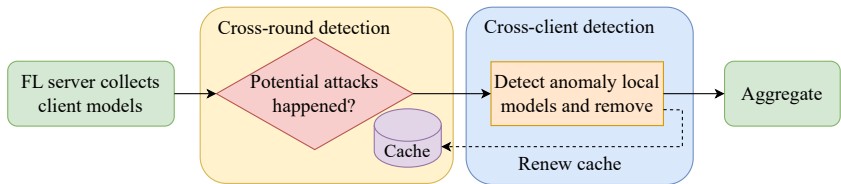

Figure 1: Overview of the proposed anomaly detection for FL systems.

Moreover, existing defense mechanisms are deployed at the FL server without any verification procedures to ensure their correct execution. While most of the clients are benign and wish to collaboratively train machine learning models, they can also be skeptical about the server's reliability due to the execution of the defense mechanisms that modify the original aggregation procedure. Thus, a successful anomaly detection approach should simultaneously satisfy the following: *i*) It should be able to detect the occurrence of attacks and exclusively handle the cases when attacks happen. *ii*) If an attack is detected, the strategy must further detect malicious client submissions and accordingly mitigate (or eliminate) their adversarial impacts without harming the benign client models. *iii*) There should be a robust mechanism to corroborate the honest execution of defense mechanisms.

In this work, we propose a novel anomaly detection mechanism that is specifically tailored to address genuine challenges faced by real-world FL systems. Our approach follows a two-stage scheme at the server to filter out malicious client submissions before aggregation. It initiates with a cross-round check based on some cache called "reference models" to determine whether any attacks have occurred. In case of attacks, a subsequent cross-client detection is executed to eliminate malicious clients models without harming the benign client models. Meanwhile, the reference models in the cache is renewed. We provide an overview in Figure 1. Our contributions are summarized as follows:

*i*) **Proactive attack detection.** Our strategy is equipped with an initial cross-round check to detect the occurrence of potential attacks, ensuring that defensive methods are only activated in response to the presence of attacks, thereby maintaining the sanctity of the process in attack-free scenarios.

*ii*) **Enhanced anomaly detection.** By coupling the cross-round check with a subsequent cross-client detection, our approach efficiently eliminates malicious client submissions without harming the benign local submissions.

*iii*) **Autonomy from prior knowledge.** Our method operates effectively without any prerequisites such as data distribution or the number of malicious clients. Such autonomous nature ensures widespread applicability and adaptability of our approach across different FL tasks, regardless of the data distribution and the selection of models.

*iv*) **Rigorous verification protocol.** Incorporating Zero-Knowledge Proof (ZKP) (Goldwasser et al., 1989) methodologies, our approach guarantees that the elimination of malicious client models is executed correctly, ensuring that clients can place trust in the defense mechanism in the FL system.

## 2 PROBLEM SETTING

### 2.1 ADVERSARY MODEL

We consider an FL system where some clients are malicious, while most clients are honest. The clients have full access to their local data and can train models using their data. They also would like to collaboratively train a model without sharing their data. However, the malicious clients among them may conduct attacks to achieve *some adversarial goals*, including: *i*) planting a backdoor to misclassify a specific set of samples while minimally impacting the overall performance of the global model, *i.e.*, backdoor attacks (Bagdasaryan et al., 2020b; Wang et al., 2020); *ii*) altering the local models to prevent the global model from converging, *i.e.*, Byzantine attacks (Chen et al., 2017; Fang et al., 2020); and *iii*) randomly submitting contrived models without actual training, *i.e.*, free riders (Wang, 2022). Clients are aware that the server can take some defensive methods to remove potential malicious local models, and they want to verify the mechanism is processed correctly and honestly at the server.

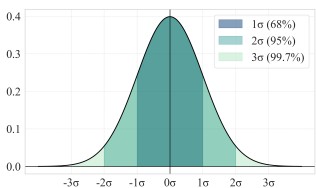 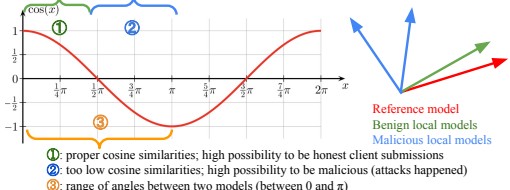

Figure 2: Three Sigma Rule.                     Figure 3: Cosine similarities.

## 2.2 PRELIMINARIES

**Krum.** Krum (Blanchard et al., 2017) is a well-known distance-based anomaly detection method in distributed learning that accepts local models that deviate less from the majority based on their pairwise distances. Given that there are $f$ byzantine clients among $L$ clients that participate in each FL iteration, Krum selects one model that is the most likely to be benign as the global model. An optimization of Krum is $m$-Krum (Blanchard et al., 2017) that selects $m$ local models, instead of one, to compute an average model when aggregating local models. Algorithms for Krum and $m$-Krum is shown in Algorithm 2 in Appendix A.1.

**Three Sigma Rule.** The three sigma rule is an empirical rule stating that almost all of the population lies within three standard deviations of the mean in normal distributions. Specifically, in normal distributions $\mathcal{N}(\mu, \sigma)$, the percentage of values within one, two, and three standard deviations of the mean are 68%, 95%, and 99.7%, respectively. This rule can be widely applied in real-world applications, since normal distributions are consistent with real-world data distributions (Lyon, 2014), and according to the central limit theorem (Rosenblatt, 1956), when aggregating independent random variables, even if the variables are generated by various distributions, the aggregation tends towards a normal distribution. Further, when the data are not normally distributed, we can transform the distribution to a normal distribution (Aoki, 1950; Osborne, 2010; Sakia, 1992; Weisberg, 2001). The three sigma rule has been used in anomaly detection (Han et al., 2019) since data outside two or three standard deviations of the mean take a very limited proportion; see Figure 2.

**Zero-knowledge proofs.** Zero-Knowledge Proofs (ZKPs)(Goldwasser et al., 1989) is a proof system that allow a prover to convince a verifier that a function on prover's secret input (witness) is correctly computed. ZKPs ensures three properties: correctness, soundness, and zero-knowledge. Correctness means that if the prover is honest, then the proof they produce should check out (integrity property). Soundness ensures that a cheating prover will not convince verifier with overwhelming probability. Zero-knowledge guarantees that prover's winess will be not learned by the verifier (privacy). Due to these properties, ZKP has been widely used in machine learning and blockchain applications (Lee et al., 2020; Feng et al., 2021; Liu et al., 2021; Sasson et al., 2014).

## 3 THE PROPOSED TWO-STAGED ANOMALY DETECTION

### 3.1 CROSS-ROUND CHECK

The goal of the cross-round check is to detect whether attacks happened in the current FL iteration. Below, we first give a high-level idea of the algorithm, then explain the algorithm in more details.

*Overview*. In the cross-round check, the server computes similarity scores between some "reference models" stored in cache and the local models of the current FL training round. As in Fung et al. (2020), we utilize cosine similarities, where lower cosine similarities indicate higher likelihood of attack occurrences. For each local model, denoted as $\mathbf{w}_i$, and a reference model, denoted as $\mathbf{w}_r$, the cosine similarity $S_c(\mathbf{w}_i, \mathbf{w}_r)$ is computed as $S_c(\mathbf{w}_i, \mathbf{w}_r) = \frac{\mathbf{w}_i \cdot \mathbf{w}_r}{||\mathbf{w}_i|| \cdot ||\mathbf{w}_r||}$. The reference models are the global model of the last training round and respective cached local models of each client from the last FL training round. Figure 3 illustrates the information deduced from the cosine similarities regarding malicious and benign local models. We expect the similarity scores across local models to be high, which shows that those local models are more likely to be benign and indicates a "converging" trend for the FL training, while lower similarities might indicate attacks

---

**Algorithm 1:** Cross-round check

---

**Inputs:** $\tau$: training round id, *e.g.*, $\tau = 0, 1, 2, \ldots$; $\mathcal{W}^{(\tau)}$: client submissions of the current FL training round; $\gamma$: upper bound of similarities for malicious client models.

1   **function** $cross\_round\_check(\mathcal{W}^{(\tau)}, \tau, \gamma)$ **begin**
2      **if** $\tau=0$ **then return** *True*;
3      $potential\_malicious\_clients \leftarrow [], attacks\_detected \leftarrow False$
4      $\mathcal{W}^{\tau-1} \leftarrow get\_cached\_client\_models()$
5      $\mathbf{w}_g^{\tau-1} \leftarrow get\_global\_model\_of\_last\_round()$
6      **for** $\mathbf{w}_i^{(\tau)} \in \mathcal{W}^\tau$ **do**
7         $S_c \leftarrow compute\_cosine\_similarity(\mathbf{w}_i^{(\tau)}, \mathbf{w}_i^{(\tau-1)})$
8         $S_c^{(g)} \leftarrow compute\_cosine\_similarity(\mathbf{w}_i^{(\tau)}, \mathbf{w}_g^{(\tau-1)})$
9         **if** $S_c < \gamma$ *or* $S_c^{(g)} < \gamma$ **then**
            $potential\_malicious\_clients.add(\mathbf{w}_i^{(\tau)})$
10           **if** $attacks\_detected$ *is False* **then** $attacks\_detected \leftarrow True$;
11      $save(potential\_malicious\_clients)$
12      **return** $attacks\_detected$

---

happened in the current training round, as the malicious clients may have submitted arbitrary or tampered local models through some attacks (Bagdasaryan et al., 2020b; Wang et al., 2020; Chen et al., 2017; Fang et al., 2020), introducing low similarities. The algorithm sets a threshold for the similarity scores for malicious models, denoted by $\gamma$. Based on this threshold, the server determines whether potential attacks have happened in the current training round.

The cross-round check algorithm is given in Algorithm 1, which has the following steps.

**Step 1: Initialization.** The server loads the reference models, including the global model from the last FL training round, as well as the cached local models that are deemed as "benign" from the previous FL training round (Line 4 and 5). For the first round that does not have a reference global model and cached local models, the algorithm assumes there are attacks and skips the cross-round check stage to directly go into the second stage.

**Step 2: Detect potential attacks.** For each local model $\mathbf{w}_i^{(\tau)}$ submitted by the $i^{th}$ client in round $\tau$, the server computes two cosine similarity scores: *i*) a cosine score $S_c$ between $\mathbf{w}_i^{(\tau)}$ and the cached local model from the same client, denoted as $\mathbf{w}_i^{(\tau-1)}$ (Line 7), and *ii*) a cosine score $S_c^{(g)}$ between $\mathbf{w}_i^{(\tau)}$ and the global model of the last FL round, denoted as $\mathbf{w}_g^{(\tau-1)}$ (Line 8). Using the two cosine similarity scores for $\mathbf{w}_i^{(\tau)}$, the algorithm detects whether potential attacks have happened in the current FL training round. To do so, we set a threshold, denoted as $\gamma$, for the cosine similarity scores, where $-1 < \gamma < 1$. Any cosine score that is lower than $\gamma$ indicates that potential attack has happened in the current FL round. Note that at this stage we do not actually remove the models that are detected malicious; instead, we just mark them as "potentially malicious" and decide whether to remove them in the latter stage of the proposed approach.

**Step 3: Return an indicator of potential attacks.** If any client models are detected as "potentially malicious", the server outputs an indicator that attacks might have happened in the current FL round, and the algorithm then enters the next stage to further inspect and remove malicious client models.

## 3.2   CROSS-CLIENT ANOMALY DETECTION

During detection, the server utilizes $3\sigma$ rule to further determine whether attacks have indeed happened on the potentially malicious clients determined in the first stage. The $3\sigma$ rule is pivotal for two reasons: *i*) parameters of local models are i.i.d. in distributed learning (Baruch et al., 2019; Chen et al., 2017; Yin et al., 2018); and *ii*) even when client data are non-i.i.d., the Central Limit Theorem (Rosenblatt, 1956) indicates that local models tend towards a normal distribution. In each FL iteration, the server computes an $\mathcal{L}_2$ score between each local model and an approximate average

model. The server then uses these scores to compute an approximate normal distribution. Based on $3\sigma$ rule, the server computes a bound for filtering out potentially malicious client models. The algorithm is in Algorithm 3 in §A.2. Below, we explain the algorithm in steps.

**Step 1: Obtain an average model.** Our algorithm takes the global model computed after removing all malicious local models in the last round as the average model for the current FL round. For the first FL training round that does not have an average model for reference, our algorithm uses $m$-Krum to compute an approximate average model. As the FL server does not know the number of potential malicious clients, we set $m$ to $L/2$ to compute an approximate average model based on the assumption that the number of malicious clients is less than $L/2$, where $L$ is the number of clients in each FL round. Such an approximate average model is used to compute $\mathcal{L}_2$ distances for local models in the current FL training round.

**Step 2: Compute scores for each client model.** The algorithm utilizes the average model, denoted as $\mathbf{w}_{avg}$, and uses it to compute an $\mathcal{L}_2$ score (*i.e.*, the Euclidean distance) for each local model $\mathbf{w}_i$ as $\mathcal{S}_i = ||\mathbf{w}_i - \mathbf{w}_{avg}||$ in the current FL training round.

**Step 3: Compute an approximate distribution of the $\mathcal{L}_2$ scores.** The server computes an approximate normal distributions using the Euclidean distance scores, specifically, the two parameters, the mean $\mu$ and the standard deviation $\sigma$. Let $\mathcal{S}_E$ denote a list of $\mathcal{L}_2$ distances with cardinality $|\mathcal{S}_E|$, then, $\mu$ and $\sigma$, are given by $\mu = \frac{\sum_{\ell \in \mathcal{S}_E} \ell}{|\mathcal{S}_E|}$ and $\sigma = \sqrt{\frac{\sum_{\ell \in \mathcal{S}_E} (\ell - \mu)^2}{|\mathcal{S}_E| - 1}}$.

**Step 4: Remove malicious local models based on the three sigma rule and the $\mathcal{L}_2$ scores.** The bound for removing malicious clients is defined as $\lambda$ ($\lambda > 0$) standard deviation of the mean as $\mu + \lambda\sigma$. The server deems local models with scores higher than the boundary as "anomaly local models" and removes them from the aggregation. Note that we only take one side of the bounds of the three sigma rule, as we prefer lower $\mathcal{L}_2$ scores, which indicate that the local model is "closer" to the average model. Thus, we do not filter out client models with scores lower than $\mu - \lambda\sigma$.

**Step 5: Compute a new average model for later FL iteration.** After removing malicious client models, the server uses the benign local models to compute an average model for the next round.

**Optimizations for computation and storage.** Algorithm 1 and Algorithm 3 utilize local models and global models to compute scores, which requires storing entire client models in cache and use them in computation. To reduce the cache size and the computation time, similar to Fung et al. (2020), we utilize a layer that can represent a whole model, called the "importance layer", instead of using full models. Intuitively, we select the second-to-the-last layer, as it contains more information in regards to the whole model. We experimentally verify this in Section 5.

## 4   Verifiable Anomaly Detection Using ZKP

This section introduces a ZKP-based verification procedure so that the benign clients can ensure the correct execution of the anomaly detection mechanism at the server. The intuition is that clients may be skeptical about the removal of some client models during the anomaly detection process at the server, as it may change the aggregation results. To verify anomaly detection procedure, we utilize zkSNARKs (Bitansky et al., 2012), which offer constant proof size and constant verification time regardless of the size of computation. This property is crucial for applications where the verifier's (*i.e.*, FL client) resources are limited. Details of implementations are deferred to Appendix §B. In what follows, the prover is the FL server, whereas the verifiers are the FL clients.

**ZKP-Compatible Language.** The first challenge of applying ZKP protocols is to convert the computations into a ZKP-compatible language. ZKP protocols model computations as arithmetic circuits with addition and multiplication gates over a prime field. However, our computations for anomaly detection are over real numbers. The second challenge is that some computations such as square root are nonlinear, making it difficult to wire them as a circuit. To address these issues, we implement a class of operations that map real numbers to fixed-point numbers. To build our ZKP scheme, we use Circom library (Contributors, 2022), which can compile the description of an arithmetic circuit in a front-end language similar to C++ to back-end ZKP protocol.

Table 1: Models and datasets for evaluations.

| Dataset | FEMNIST (Caldas et al., 2018) | Cifar10, Cifar100 (Krizhevsky et al., 2009) | Shakespeare (McMahan et al., 2017b) |
|---|---|---|---|
| Model | CNN (McMahan et al., 2017a) | ResNet20, ResNet56 (He et al., 2016) | RNN (bi-LSTM) (McMahan et al., 2017a) |

**ZKP for Anomaly Detection.** Most of the computations in Algorithm 1 and Algorithm 3 are linear, which can be compiled into an arithmetic circuit easily. For instance, computing cosine similarity between two matrices of size $n \times n$ requires a circuit with $O(n^2)$ multiplication gates and one division. While it is difficult to directly compute division on a circuit, it can be easily verified with the prover providing the pre-computed quotient and remainder beforehand. We can utilize this idea and apply Freivalds' algorithm (Freivalds, 1977) to verify matrix multiplications.

Matrix multiplication constitutes the basis of the verification schemes used for anomaly detection. Naively verifying a matrix multiplication $AB = C$ where $A, B, C$ are of size $n \times n$ requires proving the computation step by step, which requires $O(n^3)$ multiplication gates. With Freivalds' algorithm, the prover first computes the result off-circuit and commits to it. Then, the verifier generates a random vector $v$ of length $n$, and checks $A(Bv) \stackrel{?}{=} Cv$. This approach reduces the size of the circuit to $O(n^2)$. We exploit this idea of verifying the computation again to design an efficient protocol for square root, which is used in Algorithm 3. To verify that $x = \sqrt{y}$ is computed correctly, we ask the prover to provide the answer $x$ as witness and then we check in the ZKP that $x$ is indeed the square root of $y$. Note that we cannot check $x^2$ is equal to $y$ because zkSNARK works over prime field and the square root of an input number might not exist. Therefore, we check if $x^2$ is close to $y$ by checking that $x^2 \leq y$ and $(x+1)^2 \geq y$. This approach reduces the computation of square root to 2 multiplication and 2 comparison.

## 5 EVALUATIONS

This section presents a comprehensive evaluation of our approach. We focus on the following aspects: (*i*) effectiveness of cross-round check detection; (*ii*) effectiveness of the cross-client detection; (*iii*) performance comparison of our approach against other defenses; (*iv*) robustness of our approach against a dynamic subset of malicious clients; (*v*) defense coverage of our approach on different models; (*vi*) performance of our ZKP-verified anomaly detection protocol.

**Experimental setting.** A summary of datasets and models for evaluations can be found in Table 1. By default, we employ CNN and the non-i.i.d. FEMNIST dataset (partition parameter $\alpha = 0.5$), as the non-i.i.d. setting closely captures real-world scenarios. We utilize FedAVG in our experiments. By default, we use 10 clients for FL training – all clients participate in training in each round. We employ three attack mechanisms, including Byzantine attacks of random mode (Chen et al., 2017; Fang et al., 2020), and model replacement backdoor attack (Bagdasaryan et al., 2020b). We utilize three baseline defense mechanisms: $m$-Krum (Blanchard et al., 2017), Foolsgold (Fung et al., 2020), and RFA (Pillutla et al., 2022). For $m$-Krum, we set $m$ to 5, which means 5 out of 10 submitted local models participate in aggregation in each FL training round. By default, the results are evaluated with the accuracy of the global model. Evaluations for anomaly detection are conducted on a server with 8 NVIDIA A100-SXM4-80GB GPUs, and evaluations for ZKP are conducted on Amazon AWS with an m5a.4xlarge instance with 16 CPU cores and 32 GB memory.

**Exp1: Selection of importance layer.** We utilize the norm of gradients to evaluate the "sensitivity" of each layer. A layer with a norm higher than most of the other layers indicates higher sensitivity compared with most of the other layers, thus can be utilized to represent the whole model. We evaluate the sensitivity of layers of CNN, RNN, and ResNet56. The results for RNN are shown in Figure 4, and the results for ResNet56 and CNN are deferred to Figure 14 and Figure 13 in Appendix §B.2, respectively. The results show the sensitivity of the second-to-the-last layer is always higher than most of the other layers, which includes adequate information of the whole model, thus can be selected as the importance layer.

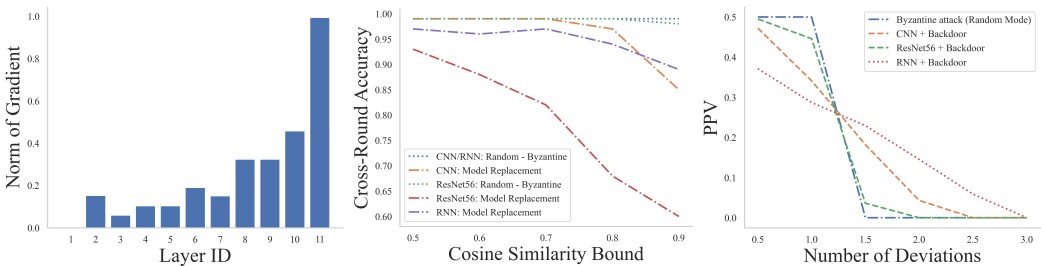

Figure 4: RNN layer sensitivity.  Figure 5: Varying $\gamma$.  Figure 6: Varying # deviations.

## 5.1 CROSS-ROUND CHECK EVALUATIONS

In this subsection, we assess the efficiency of the cross-round check in detecting attacks. We employ the random-Byzantine and the model replacement backdoor attacks, and set the attack probability to 40% for each FL iteration, where 1 out of the clients are malicious when the attack happens. Ideally, the cross-round check should confirm the absence or presence of an attack accurately. The effectiveness of our approach is evaluated by the cross-round check accuracy, which measures the proportion of iterations in which the algorithm correctly detects cases with or without an attack relative to the number of total iterations. A 100% cross-round check accuracy means that all attacks are detected, and none of the benign cases are identified as "attacks".

**Exp2: Impact of the similarity threshold.** We evaluate the impact of the cosine similarity threshold $\gamma$ in the cross-round check by setting $\gamma$ to 0.5, 0.6, 0.7, 0.8, and 0.9. According to Algorithm 1, a cosine score lower than $\gamma$ indicates lower similarities between client models of the current round and the last round, which may be an indicator of the occurance of an attack. As shown in Figure 5, the cross-round accuracy is close to 100% in the case of random-Byzantine attacks. Further, when the cosine similarity threshold $\gamma$ is set to 0.5, the performance is good in all cases, with at least 93% cross-round detection accuracy.

## 5.2 CROSS-CLIENT DETECTION EVALUATIONS

This subsection evaluates whether the cross-client detection can successfully detect malicious client submissions given the cases with attacks. We evaluate the quality of anomaly detection using modified Positive Predictive Values (PPV) (Fletcher, 2019), *i.e.,* precision, the proportions of positive results in statistics and diagnostic tests that are true positive results. Let us denote the number of true positive and the false positive results to be $N_{TP}$ and $N_{FP}$, respectively, then $PPV = \frac{N_{TP}}{N_{TP}+N_{FP}}$. In the context of anomaly detection in FL, client submissions that are detected as "malicious" and are actually malicious are defined as *True Positive*, corresponding to $N_{TP}$, while client submissions that are detected as "malicious" even though they are benign are defined as *False Positive*, corresponding to $N_{FP}$. Since we would like the PPV to reveal the relation between $N_{TP}$ and the total number of malicious submissions across all FL iterations, denoted as $N_{total}$, whether they are detected or not, we introduce a modified PPV as $PPV = \frac{N_{TP}}{N_{TP}+N_{FP}+N_{total}}$, where $0 \leq PPV \leq \frac{1}{2}$. In ideal cases, all malicious submissions are detected, where $N_{TP} = N_{total}$, and $PPV$ is $\frac{1}{2}$. Due to the page limitations we deferred the proof of this statement to Appendix A.3.

**Exp 3: Selection of the number of deviations.** This experiment utilizes $PPV$ to evaluate the impact of the number of deviations, *i.e.*, the parameter $\lambda$ in the anomaly bound $\mu + \lambda\sigma$. To evaluate a challenging case where a large portion of the clients are malicious, we set 4 out of 10 clients to be malicious in each FL training round. Given the number of FL iterations to be 100, the total number of malicious submissions $N_{total}$ is 400. We set the number of deviations $\lambda$ to be 0.5, 1, 1.5, 2, 2.5, and 3. We select the Byzantine attack of random mode and the model replacement attack, and evaluate our approach on three tasks, as follows: *i)* CNN+FEMNIST, *ii)* ResNet-56+Cifar100, and *iii)* RNN + Shakespeare. The results, as shown in Figure 6, indicate that when $\lambda$ is 0.5, the results are the best, with the $PPV$ being at least 0.37. In fact, when $\lambda$ is 0.5, all malicious submissions are detected for the random Byzantine attack for all three tasks, with the $PPV$ being exactly 1/2. In subsequent experiments, unless specified otherwise, we set $\lambda$ to 0.5 by default.

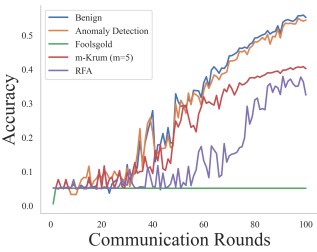
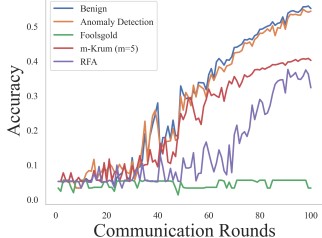
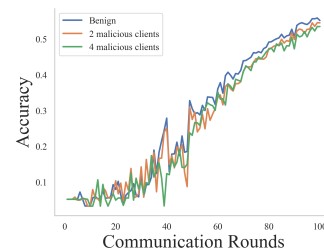

Figure 7: Defenses against random-Byzantine attack.

Figure 8: Defenses against model replacement attack.

Figure 9: Varying # malicious clients.

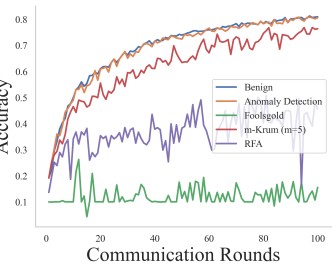
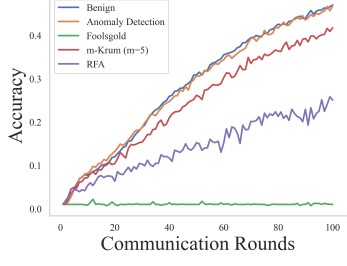
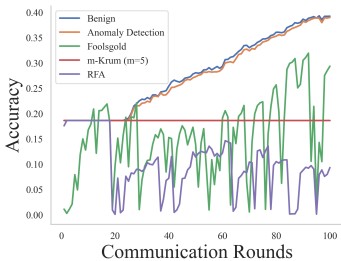

Figure 10: ResNet20 & Cifar10  Figure 11: ResNet56 & Cifar100  Figure 12: RNN & Shakespeare

**Exp 4: Comparisons between anomaly detection and various defenses against attacks.** This experiment evaluates the effect of our approach compared with various defense mechanisms, including Foolsgold, $m$-Krum ($m = 5$), and RFA in the context of ongoing attacks. We include a "benign" baseline scenario with no activated attack or defense, and select random-Byzantine and model replacement backdoor attacks. The results for the random-Byzantine attack and model replacement backdoor attack are shown in Figure 7 and Figure 8, respectively. These results demonstrate that our approach effectively mitigates the negative impact of the attacks and significantly outperforms the other defenses, with the test accuracy much closer to the benign case.

**Exp 5: Varying the number of malicious clients.** This experiment evaluates the impact of varying number of malicious clients on test accuracy. We set the number of malicious clients out of 10 clients in each FL training round to 2 and 4, and include a baseline case where all clients are benign. As shown in Figure 9, the test accuracy remains relatively consistent across different numbers of malicious clients, as in each FL training round, our approach filters out the local models that tend to be malicious to effectively minimize the impact of malicious client models on the aggregation.

**Exp 6: Evaluations on different models.** We evaluate defense mechanisms against the random mode of the Byzantine attack with different models and datasets, including: *i*) ResNet-20 + Cifar10, *ii*) ResNet-56 + Cifar100, and *iii*) RNN + Shakespeare. The results are shown in Figures 10, 11, and 12, respectively. The results show that while the baseline defense mechanisms can mitigate the impact of attacks in most cases, some defenses may fail in some tasks, *e.g.*, $m$-Krum fails in RNN in Figure 12. This is because those methods either select a fixed number of local models or re-weight the local models in aggregation, which potentially eliminates some local models that are important to the aggregation, leading to an unchanged test accuracy in later FL iterations. In contrast, our approach outperforms those baseline defense methods by effectively filtering out local models that are detected as "outliers", with a test accuracy close to the benign scenarios.

### 5.3  ZKP Performance

**Exp 7: Evaluations of ZKP circuit size, proving time, and verification time.** We implement the ZKP system in Circom (Contributors, 2022). This system includes a module for cross-round check and cross-client detection. We also implement a prover's module which contains JavaScript code to generate witness for the ZKP, as well as to perform fixed-point quantization. In our experiments, we include CNN, RNN, and ResNet-56 as our machine learning models. Specifically, we only pull out

parameters from the second-to-last layer of each model, *i.e.*, the importance layer, as our weights to reduce complexity. For instance, the second-to-last layer of the CNN model contains only $7,936$ trainable parameters, as opposed to $1,199,882$ should we use the entire model. We implement both algorithms across 10 clients, and report our results in Table 2.

Table 2: Cost of ZKP in various models.

| Model | Stage 1 Circuit Size | Stage 2 Circuit Size | Proving (s) | Verification (ms) |
|---|---|---|---|---|
| CNN | 476,160 | 795,941 | 33 (12 + 21) | 3 |
| RNN | 1,382,400 | 2,306,341 | 96 (34 + 62) | 3 |
| ResNet-56 | 1,536,000 | 2,562,340 | 100 (37 + 63) | 3 |

Note: Bracketed times denote durations for Stage 1 (Cross-round) and Stage 2 (Cross-client).

## 6 RELATED WORKS

**Detection of occurrence of attacks.** Zhang et al. (2022b) employs $k$-means to partition local models into clusters that correspond to "benign" or "malicious". While this approach can efficiently detect potential malicious client models, it relies too much on historical client models from previous training rounds and might not be as effective when there is limited information on past client models. For example, in their implementation (Zhang et al., 2022a), since they need to collect historical client model information, authors set the starting round to detect attacks to different training rounds, *e.g.*, 50 when the datasets are MNIST and FEMNIST, and 20 when the dataset is CIFAR10. Apparently, this is not suitable for real FL systems, as attacks may happen in earlier rounds as well.

**Defense mechanisms in FL.** Robust learning and the mitigation of adversarial behaviors in FL has been extensively explored (Blanchard et al., 2017; Yang et al., 2019; Fung et al., 2020; Pillutla et al., 2022; He et al., 2022; Karimireddy et al., 2020; Sun et al., 2019; Fu et al., 2019; Ozdayi et al., 2021; Sun et al., 2021; Yin et al., 2018; Chen et al., 2017; Guerraoui et al., 2018; Xie et al., 2020; Li et al., 2020; Cao et al., 2020). Some approaches keep several local models that are more likely to be benign in each FL iteration, *e.g.*, (Blanchard et al., 2017; Guerraoui et al., 2018; Yin et al., 2018), and (Xie et al., 2020). For each FL round, instead of using all client submissions for aggregation, such approaches keep local models that are the most likely to be benign to represent the other local models. Such approaches are effective, but they keep less local models than the real number of benign local models to ensure that all Byzantine local models are filtered out, causing missing representation of some benign local models in aggregation. This completely wastes the computation resources of the benign clients that are not being selected and thus, changes the aggregation results as some benign local models do not participate in aggregation. Some approaches re-weight or modify local models to mitigate the impacts of potential malicious submissions (Fung et al., 2020; Karimireddy et al., 2020; Sun et al., 2019; Fu et al., 2019; Ozdayi et al., 2021; Sun et al., 2021), while other approaches alter the aggregation function or directly modify the aggregation results (Pillutla et al., 2022; Karimireddy et al., 2020; Yin et al., 2018; Chen et al., 2017). While these defense mechanisms can be effective against attacks, they might inadvertently degrade the quality of outcomes due to the unintentional alteration of aggregation results even when no attacks are present. This is especially problematic given the infrequency of attacks in real-world FL scenarios.

## 7 CONCLUSION

We present a cutting-edge anomaly detection technique specifically designed for the real-world FL systems. Our approach utilizes an early cross-round check that activates subsequent anomaly detection exclusively in the presence of attacks. When attacks happen, our approach removes anomaly client models efficiently, ensuring that the local models submitted by benign clients remain unaffected. Further, by leveraging ZKPs, participating clients get to verify the integrity of the anomaly detection and removal performed by the server. The practical design and the inherent efficiency of our approach make it well-suited for real-world FL systems. Our plans for future works include extending our approach to asynchronous FL and vertical FL scenarios.

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

# A APPENDIX

## A.1 ALGORITHM FOR KRUM AND $m$-KRUM

---

**Algorithm 2:** Krum and $m$-Krum.

---

**Inputs:** $\mathcal{W}$: client submissions of a training round; $i$: the client id for which we compute a Krum score $S_K(\mathbf{w}_i)$; $f$: the number of malicious clients in each FL iteration; $m$: the number of "neighbor" client models that participate in computing the Krum score $S_k(\mathbf{w}_i)$ of each client model $\mathbf{w}_i$; $m$ is 1 by default in Krum.

1 **function** $Krum\_and\_m\_Krum(\mathcal{W}, m, f)$ **begin**
2    $S_k \leftarrow []$
3    **for** $\mathbf{w}_j \in \mathcal{W}$ **do**
4       $S_k(\mathbf{w}_i) \leftarrow compute\_krum\_score(\mathcal{W}, i, m, f)$
5    $filter(\mathcal{W}, S_k)$            $\triangleright$ Keep local models with the $L/2$ lowest Krum scores
6    **return** $average(\mathcal{W})$

7 **function** $compute\_krum\_score(\mathcal{W}, i, m, f)$ **begin**
8    $d \leftarrow []$            $\triangleright$ Square distances of $\mathbf{w}_i$ to other local models.
9    $L \leftarrow |\mathcal{W}|$          $\triangleright$ $L$: the number of clients in each FL round.
10    **for** $\mathbf{w}_j \in \mathcal{W}$ **do**
11       **if** $i \neq j$ **then**
         $d.append(||\mathbf{w}_i - \mathbf{w}_j||^2)$
12    $sort(d)$                           $\triangleright$ In ascending order
13    $S_k(\mathbf{w}_i) \leftarrow \sum_{k=0}^{L-f-3} d$    $\triangleright$ Use the smallest $L - f - 2$ scores to compute $S_k(\mathbf{w}_i)$
14    **return** $S_k(\mathbf{w}_i)$

---

## A.2 ANOMALY DETECTION ALGORITHM

## A.3 PROOF OF THE RANGE OF $PPV$

Below we prove the upper bound of $PPV$ to be $\frac{1}{2}$.

*Proof.* $PPV = \frac{N_{TP}}{N_{TP} + N_{FP} + N_{total}}$, then $\frac{1}{PPV} = 1 + \frac{N_{FP}}{N_{TP}} + \frac{N_{total}}{N_{TP}}$. As $\frac{N_{FP}}{N_{TP}} \geq 0$ and $\frac{N_{total}}{N_{TP}} \geq 1$, we have $\frac{1}{PPV} \geq 2$, thus $PPV \leq \frac{1}{2}$.    $\square$

---

**Algorithm 3:** Cross-client Detection Algorithm.

---

**Inputs:** $\tau$: training round id, $e.g.$, $\tau = 0, 1, 2, \ldots$; $\mathcal{W}$: client submissions of a training round.

1 **function** $Detect\_Anomaly\_Submissions(\mathcal{W}, \tau)$ **begin**

2     **if** $\tau = 0$ **then**
        $m \leftarrow |\mathcal{W}|/2$, $f \leftarrow |\mathcal{W}|/2$, $\mathbf{w}_{\text{avg}} \leftarrow Krum\_and\_m\_Krum(\mathcal{W}, m, f)$

3     $\mathcal{S} \leftarrow compute\_L2\_scores(\mathcal{W}, \mathbf{w}_{\text{avg}})$

4     $\mu \leftarrow \frac{\sum_{\ell \in \mathcal{S}} \ell}{|\mathcal{S}|}$, $\sigma \leftarrow \sqrt{\frac{\sum_{\ell \in \mathcal{S}} (\ell - \mu)^2}{|\mathcal{S}| - 1}}$.

5     **for** $0 < i < |\mathcal{S}|$ **do**

6         **if** $\mathcal{S}[i] > \mu + \lambda\sigma$ **then** remove $\mathbf{w}_i$ from $\mathcal{W}$ ;

7     $\mathbf{w}_{\text{avg}} \leftarrow average(\mathcal{W})$

8     **return** $\mathcal{W}$

---

# B    ZKP IMPLEMENTATION

## B.1    CHOICE OF THE ZKP SYSTEM

In our implementation, we use the Groth16 (Groth, 2016) zkSNARK scheme implemented in the Circom library (Contributors, 2022) for all the computations described above. We chose this ZKP scheme because its construction ensures constant proof size (128 bytes) and constant verification time. Because of this, Groth16 is popular for blockchain applications due to small on-chain computation. There are other ZKP schemes based on different constructions that can achieve faster prover time (Liu et al., 2021), but their proof size is too big and verification time is not constant, which is a problem if the verifier lacks computational power. The construction of a ZKP scheme that is efficient for both prover and verifier is still an open research direction.

## B.2    SUPPLEMENTARY EXPERIMENTS FOR IMPORTANCE LAYER

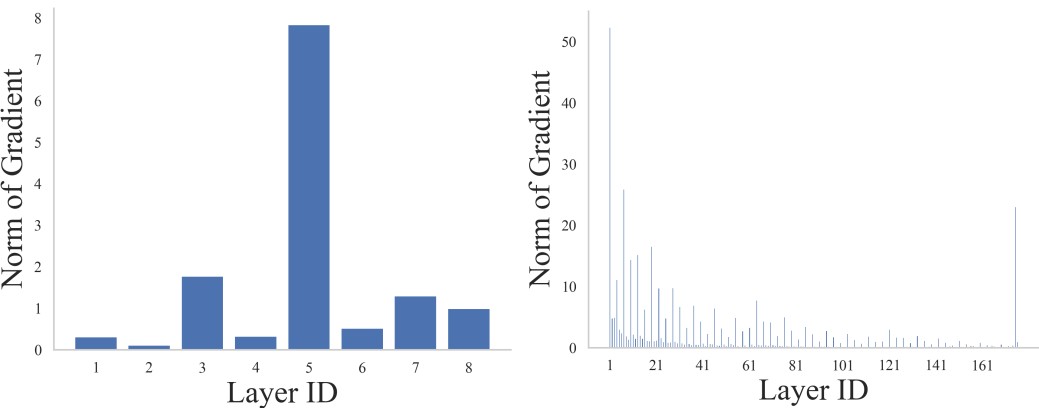

Figure 13: CNN layer sensitivity.          Figure 14: ResNet56 layer sensitivity.

## B.3    SUPPLEMENTARY EXPERIMENTS FOR REAL-WORLD APPLICATIONS

**Exp8: Evaluations in real-world applications.** We utilize edge devices from the Theta network (Theta Network., 2023) to validate the scalability of our anomaly detection approach to real-world applications. The FL client package is integrated into Theta's edge nodes, which periodically fetch data from the Theta back-end. Subsequently, the FL training server capitalizes on these Theta edge nodes and their associated data to train, fine-tune, and deploy machine learning models.

We utilize the Byzantine attack of random mode. Considering the challenges posed by real-world environments, such as devices equipped solely with CPUs (lacking GPUs), potential device con-

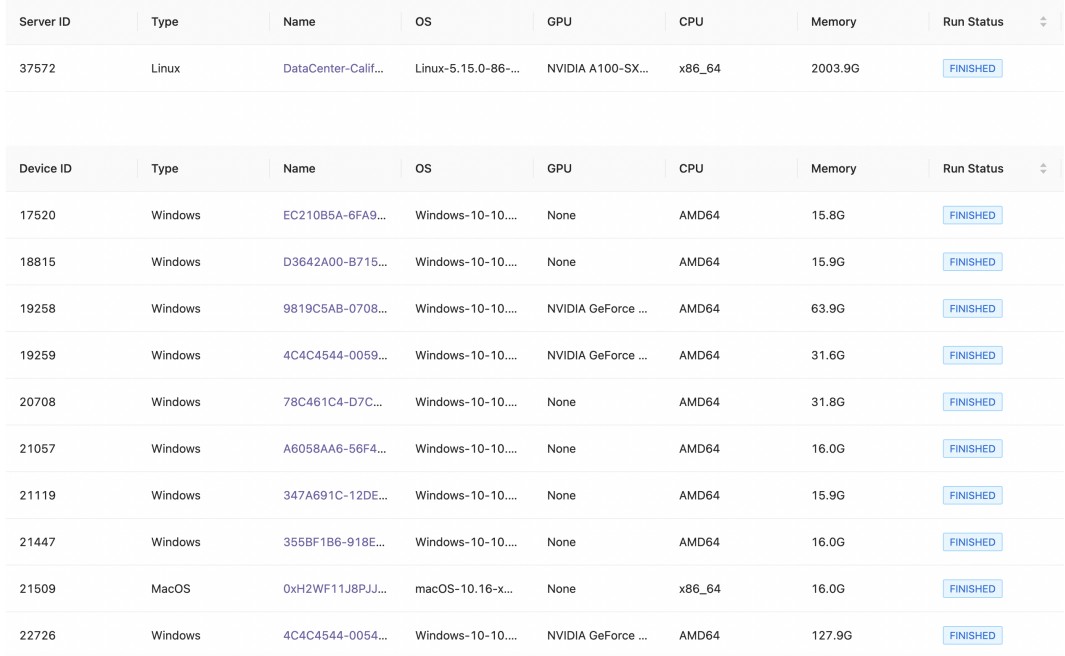

| Server ID | Type | Name | OS | GPU | CPU | Memory | Run Status |
|-----------|------|------|-----|-----|-----|--------|------------|
| 37572 | Linux | DataCenter-Calif... | Linux-5.15.0-86-... | NVIDIA A100-SX... | x86_64 | 2003.9G | FINISHED |

| Device ID | Type | Name | OS | GPU | CPU | Memory | Run Status |
|-----------|------|------|-----|-----|-----|--------|------------|
| 17520 | Windows | EC210B5A-6FA9... | Windows-10-10.... | None | AMD64 | 15.8G | FINISHED |
| 18815 | Windows | D3642A00-B715... | Windows-10-10.... | None | AMD64 | 15.9G | FINISHED |
| 19258 | Windows | 9819C5AB-0708... | Windows-10-10.... | NVIDIA GeForce ... | AMD64 | 63.9G | FINISHED |
| 19259 | Windows | 4C4C4544-0059... | Windows-10-10.... | NVIDIA GeForce ... | AMD64 | 31.6G | FINISHED |
| 20708 | Windows | 78C461C4-D7C... | Windows-10-10.... | None | AMD64 | 31.8G | FINISHED |
| 21057 | Windows | A6058AA6-56F4... | Windows-10-10.... | None | AMD64 | 16.0G | FINISHED |
| 21119 | Windows | 347A691C-12DE... | Windows-10-10.... | None | AMD64 | 15.9G | FINISHED |
| 21447 | Windows | 355BF1B6-918E... | Windows-10-10.... | None | AMD64 | 16.0G | FINISHED |
| 21509 | MacOS | 0xH2WF11J8PJJ... | macOS-10.16-x... | None | x86_64 | 16.0G | FINISHED |
| 22726 | Windows | 4C4C4544-0054... | Windows-10-10.... | NVIDIA GeForce ... | AMD64 | 127.9G | FINISHED |

Figure 15: Theta edge devices

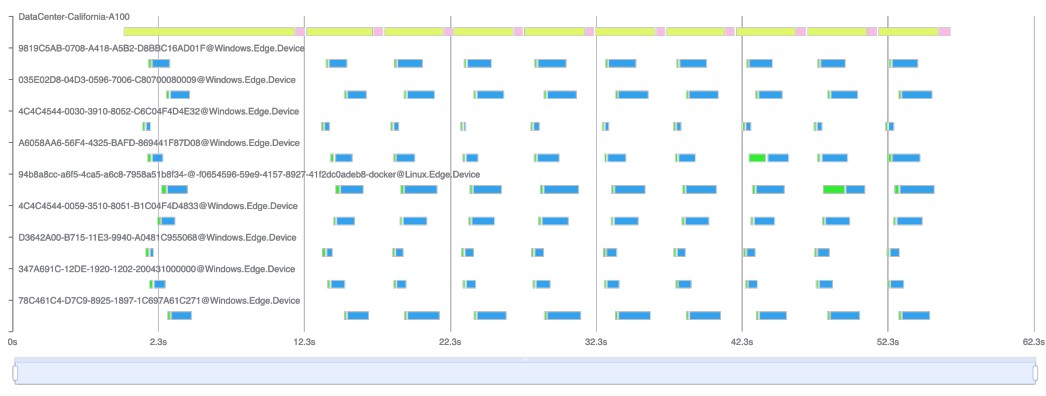

Figure 16: Real-world application on Theta network. Yellow: aggregation server waiting time; pink: aggregation time; green: client training time; blue: client communication.

nectivity issues, network latency, and limited storage on edge devices (for instance, some mobile devices might have less than 500MB of available storage), we choose a simple task by employing the MNIST dataset for a logistic regression task. We deploy 20 client edge devices, and set 5 of them as malicious for each FL training round. The information of the Theta devices is shown in Figure 15. The training process is shown in Figure 16, and the total training time is 221 seconds. We also include the CPU utilization and network traffic during training, which are shown in Figure 17 and Figure 18, respectively.

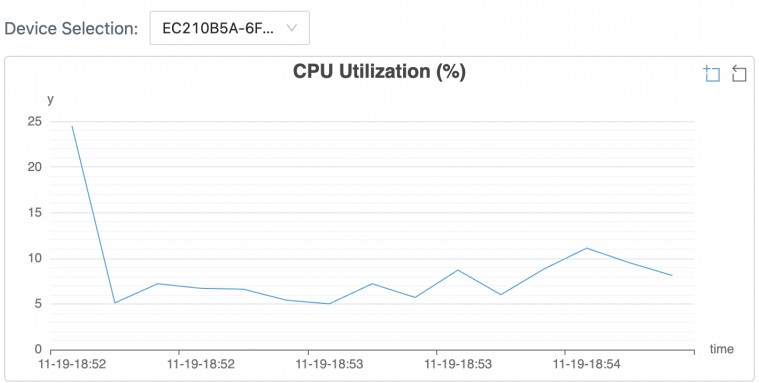

Figure 17: CPU utilization

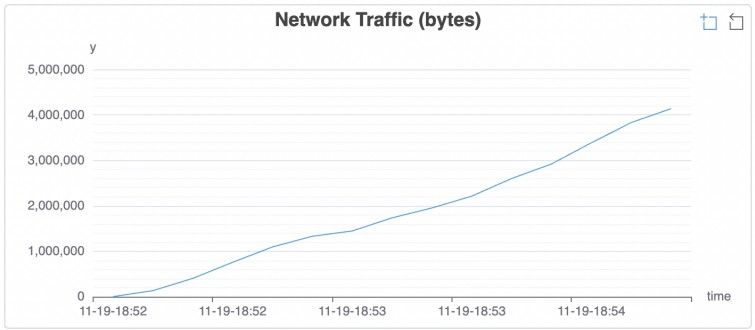

Figure 18: Network traffics

