# OpenReview forum: "Kick Bad Guys Out! Zero-Knowledge-Proof-Based Anomaly Detection in Federated Learning"
_ICLR.cc/2024/Conference — Submitted to ICLR 2024_

### Official Review · Reviewer_TEpF · 2023-10-27

**Soundness:** 2 fair
**Presentation:** 2 fair
**Contribution:** 2 fair
**Rating:** 3
**Confidence:** 4

**Summary:**

The paper proposed a FL anomaly detection method with the following features: i) Detecting the occurrence of attacks and performing defense operations only when attacks happen; ii) Upon the occurrence of an attack, further detecting the malicious client models and eliminating them without harming the benign ones; iii) Ensuring honest execution of defense mechanisms at the server by leveraging a zero-knowledge-proof mechanism.

**Strengths:**

1. The authors proposed a new method for FL anomaly detection.

**Weaknesses:**

1. The authors did not compare with FL anomaly detection methods such as FLDetector in experiments.
2. The authors did not test their methods with some strong attacks such as [1], [2] , and [3].

[1] Gilad Baruch, Moran Baruch, and Yoav Goldberg. 2019. A Little Is Enough: Circumventing Defenses For Distributed Learning. In NeurIPS
[2] Eugene Bagdasaryan, Andreas Veit, Yiqing Hua, Deborah Estrin, and Vitaly Shmatikov. 2020. How to backdoor federated learning. In AISTATS
[3] Chulin Xie, Keli Huang, Pin-Yu Chen, and Bo Li. 2019. Dba: Distributed backdoor attacks against federated learning. In ICLR

3. The authors did not explore the influence of the number of malicious clients.

**Questions:**

Please see the weakness.

---

> ### Author Response · Authors · 2023-11-20
> **Responses to Chairs and Reviewers**
>
> Dear Chairs and Reviewers,
>
> We do appreciate your effort in reviewing our paper. We got an overall rating of 3 (reject) and a confidence of 4 (confident in their assessment) from Reviewer TEpF, however, we would like to point out that the review is problematic and full of mistakes contradicting our work. In what follows, we expand our concerns in the following aspects:
>
> ## Nearly copy-paste summary
>
> Reviewer TEpF summarized the paper as follows:
>
> ```
> The paper proposed a FL anomaly detection method with the following features: i) Detecting the occurrence of attacks and performing defense operations only when attacks happen; ii) Upon the occurrence of an attack, further detecting the malicious client models and eliminating them without harming the benign ones; iii) Ensuring honest execution of defense mechanisms at the server by leveraging a zero-knowledge-proof mechanism.
> ```
>
> While, in the abstract of our submission, we wrote:
>
> ```
> To address these challenges in real FL systems, this paper introduces a cutting-edge anomaly detection approach with the following features: i) Detecting the occurrence of attacks and performing defense operations only when attacks happen; ii) Upon the occurrence of an attack, further detecting the malicious client models and eliminating them without harming the benign ones; iii) Ensuring honest execution of defense mechanisms at the server by leveraging a zero-knowledge proof mechanism.
> ```
>
> By comparing our original text with Reviewer TEpF’s summary, we can easily see that Reviewer TEpF almost copied-pasted part of our abstract with little modification. This makes us think that the Reviewer TEpF has spent little to no time reviewing our paper, as evidenced by their other comments, which we discuss next.
>
>
> ## Comments with factual mistakes regarding a “missing” experiment
>
> In Weakness 3 (W3), the reviewer said “The authors did not explore the influence of the number of malicious clients.” However, in the original submitted version, we **exactly included** the experiment as Exp 5 (Varying the number of malicious clients) in the experiment section, – not the appendix. We varied the percentage of malicious clients to be 0, 20%, and 40%, and our approach works pretty well, – in both cases of 20% and 40% malicious clients, the test accuracy remains very close to the benign case.
>
>
> ## Controversial comments with factual mistakes regarding general experiments
>
> In W2, the reviewer said “The authors did not test their approach with some strong attacks” and then the reviewer listed 3 papers ([1][2][3]). However, we have already utilized the attack in [2] in our experiments. Moreover, [1] is in fact a famous defense paper, – even its title clearly reflects that it is a defense paper. Though that paper includes an attack, it is impractical in real-world scenarios (in the next section, we describe why it is impractical). Based on our survey so far, in our humble opinion, we have not seen any paper that utilizes the attack described in [1]. On the other hand, in our experiments, we have already utilized attacks that are both practical and stronger than the attacks suggested by the reviewer. We will explain them in detail in the following section.

---

> ### Author Response · Authors · 2023-11-20
> **Responses to Chairs and Reviewers and Detailed Responses to Reviewer TEpF**
>
> As we responded to W3 in the section above, we would like to address the remaining 2 weaknesses mentioned by Reviewer TEpF.
>
> ### W1: The authors did not compare with FL anomaly detection methods such as FLDetector in experiments.
>
> As already mentioned in our paper, FLDetector is not practical in real-world scenarios. It relies heavily on historical client models from previous training rounds. By examining their detailed implementation, we found that the number of pre-training rounds may vary based on the datasets. For example, the number of pre-training rounds is set to 50 when the datasets are MNIST and FEMNIST, and 20 when the dataset is CIFAR10. Since our approach starts with zero pre-training rounds, it is more generalized, and we do not think there is need in the experiment section to compare our approach with FLDetector which has the practical design limitations above. Thus, in our paper, we decided to mention this work in the Related Works section.
>
> ### W2: The authors did not test their methods with some strong attacks
>
> Reviewer TEpF suggested 3 attacks (paper [1][2][3]) to us. While we have already included one of the three attacks in our experiments ([2]), we would like to point out that our work has already leveraged stronger and more practical attacks in our paper, compared to those suggested by Reviewer TEpF.
>
> As addressed in the whole paper, our anomaly detection solution handles the challenges faced by real-world FL systems. The basic rule to select attacks in our experiments is that the attack must be practical to be utilized by malicious clients in real-world cases.
>
> As for the other two attacks suggested by Reviewer TEpF, [1] is a famous defense paper in fact, – even its title clearly reflects that it is a defense paper. Though they design an attack, to the best of our knowledge, there is neither an existing work leveraging this attack nor it is practical in real-world scenarios. As shown in their open-sourced implementation (https://github.com/moranant/attacking_distributed_learning/blob/master/malicious.py), the method requires the malicious clients to modify their local models based on the “distribution” of all local models in the current FL training round, using a mean and a standard deviation. Implementing this in real-world FL applications requires the server (or a malicious third party) to compute the real mean (i.e., the averaged model) and the standard deviation, then send them back to the malicious clients for them to compute the poisoned local model. If the mean (averaged model) is computed by the server, the server has no reason to take the poisoned local model again from the local clients, as it has already obtained the averaged model for the current FL round. If the mean is computed by a malicious third party, then the assumption is that the third party attacker has taken control over the whole FL system, which is too strong and almost impractical in real-world scenarios (remember we have an honest majority).
>
> Regarding the attack in [3], in fact, in our experiments, we have already considered two practical attacks ([2] in the reviewer’s suggested attacks and [4]) that are stronger than [3]. The stronger attack in our experiments can even induce the global model to be the local model submitted by a malicious client. Even under this scenario, we show that our proposed method works effectively.
>
> [4] Chen, Yudong, Lili Su, and Jiaming Xu. "Distributed statistical machine learning in adversarial settings: Byzantine gradient descent." Proceedings of the ACM on Measurement and Analysis of Computing Systems 1.2 (2017): 1-25.

---

> ### Comment · Reviewer_TEpF · 2023-11-23
> **Response to the rebuttal**
>
> Dear Chair, reviewers, and authors
>
> Thanks to the authors for the detailed reply! However, most of my concerns still exist.
>
> 1. The authors still have not included FLDetector in experiments, which is a popular method for FL anomaly detection. The authors just said the historical information is required in FLDetector, and then the method is impractical. This is not quite convincing.
>
> 2. [1] and [3] are still not compared. [1] A Little Is Enough: Circumventing Defenses For Distributed Learning is an attack paper, even from the title.
>
> 3. Systematic exploration of the influence of the number of malicious clients is expected. In Exp5, only 2 and 4 malicious clients for a single attack are considered.
>
> Therefore, I would like to keep my score. Finally, I suggest paying more attention to the paper itself, which will be more helpful.

---

> > ### Author Response · Authors · 2023-11-23
> >
> > We thank the reviewer’s timely feedback. We would like to first mention that in our previous response, we pointed out three mistakes that are easy to see in the review. However, the reviewer still argued for one mistake he/she made.
> >
> > In the reviewer’s first review, the reviewer mentioned “The authors did not explore the influence of the number of malicious clients”, which indicates he/she even did not see the experiment completely. While in his/her new review, he/she mentioned: “(W3) **Systematic exploration** of the influence of the number of malicious clients is expected”.
> >
> > Below are our responses for having not **systematically** explored the influence of the number of malicious clients.
> >
> > We set the percentage of malicious clients to 20% and 40%, and since we utilize 10 clients, the number of malicious clients is 2 and 4. As a comparison, we also included a benign case where there is no malicious client. The reason for this setting is as follows. 1) the number of clients to be 10: In real-world cases, the number of clients can hardly larger than 20, while in most cases, the number of clients is less than 10. 2) the percentage of malicious clients: we would like to first point out that security solutions often address scenarios that are less frequently encountered in the real world – those that may rarely occur, but when they do, they can lead to serious consequences. This holds true for security issues in FL systems as well. In real FL systems, the proportion of malicious clients is typically no more than 10%. We would like to evaluate the performance of our approach to handle more malicious clients, thus we set the percentage of malicious clients to be 20% and 40%. The results show that our approach works pretty well, – in both cases of 20% and 40% malicious clients, the test accuracy remains very close to the benign case. 3) The reason why we did not include 10% and 30%: the performance of our approach is quite good, and the test accuracy always remains very close to the benign case when we vary the percentage of malicious clients from 10% to 40%. To make the lines in the figure clear and easy to see, we decided to only include two settings and set the percentage of malicious clients to 20% and 40% only.
> >
> > Below are our responses for the reviewer’s other concerns.
> >
> > W1: We acknowledge that FLDetector is a good paper. However, there is often a gap between “a good paper” and “practical in real industry products”. For example, a research paper may assume attacks always happen, while in real systems, attacks hardly happen, and the adversary definitely would not notify users or system owners before attacking. But we cannot say those works are not important. Back to your concern, we have clearly mentioned why we did not include it as a comparison in experiments in our previous response, as follows.
> >
> > ```
> > As already mentioned in our paper, FLDetector is not practical in real-world scenarios. It relies heavily on historical client models from previous training rounds. By examining their detailed implementation, we found that the number of pre-training rounds may vary based on the datasets. For example, the number of pre-training rounds is set to 50 when the datasets are MNIST and FEMNIST, and 20 when the dataset is CIFAR10. Since our approach starts with zero pre-training rounds, it is more generalized, and we do not think there is a need in the experiment section to compare our approach with FLDetector which has the practical design limitations above. Thus, in our paper, we decided to mention this work in the Related Works section.
> > ```
> > If it is still not convincing enough, we would like to explain more about the details of FLDetector.  This method utilizes lbfgs to calculate hessian vector product (hvp), which is used to detect whether attacks happened later. However, they set different start rounds for different datasets/tasks to compute hpv. We include their implementation for reference, as follows:
> >
> > 1. In Line 455 of their code for Cifar experiments (https://github.com/zaixizhang/FLDetector/blob/main/train_cifar.py), they set the start round to be 20.
> >
> > 2. In Line 382 of their Feminist experiment implementation (https://github.com/zaixizhang/FLDetector/blob/main/train_femnist.py), they set the start round to 50.
> >
> > 3. In Line 465 of their Mnist experiment implementation (https://github.com/zaixizhang/FLDetector/blob/main/train_mnist.py), they set the start round to be 50.
> >
> > The reviewer should know that fair comparison is the top 1 requirement in research experiments. While our approach does not require any pre-processing FL iterations, we don’t think it’s fair to compare our method by modifying their methods to remove their historical model loading process.

---

> > ### Author Response · Authors · 2023-11-23
> >
> > W2: We would like to restate our reasons why we did not consider the attack in [1]. To the best of our knowledge, it is not practical in real-world scenarios and is not widely considered in literature. As shown in their open-sourced implementation (https://github.com/moranant/attacking_distributed_learning/blob/master/malicious.py), the method requires the malicious clients to modify their local models based on the “distribution” of all local models in the current FL training round, using a mean and a standard deviation. Implementing this in real-world FL applications requires the server (or a malicious third party) to compute the real mean (i.e., the averaged model) and the standard deviation, then send them back to the malicious clients for them to compute the poisoned local model. If the mean (averaged model) is computed by the server, the server has no reason to take a poisoned local model again from the local clients, as it has already obtained the averaged model for the current FL round. If the mean is computed by a malicious third party, then the assumption is that the third party attacker has taken control over the whole FL system, which is too strong and almost impractical in real-world scenarios.
> > Regarding the attack in [3], we still would like to restate the reason why we did not include it in our experiments. In fact, [3] proposes a backdoor attack, and in our experiments, we have already considered a backdoor attack (i.e., [2]) that is more practical and stronger than [3]. The stronger attack in our experiments can even induce the global model to be the local model submitted by a malicious client. Even under this scenario, we show that our proposed method works effectively.

---

### Official Review · Reviewer_zU9y · 2023-10-28

**Soundness:** 2 fair
**Presentation:** 3 good
**Contribution:** 2 fair
**Rating:** 5
**Confidence:** 3

**Summary:**

The paper introduces a novel anomaly detection approach for addressing the vulnerability of federated learning (FL) systems to malicious clients. It focuses on detecting and eliminating malicious client models without harming benign ones, using a zero-knowledge proof mechanism to ensure honest execution of defense mechanisms at the server.

**Strengths:**

The proposed approach detects attacks and performs defense operations only when attacks occur, and further identifies and removes malicious client models, therefore being harmless on benign ones.

**Weaknesses:**

1. The mechanism relies heavily on the assumption that malicious clients will remain below 50% of the total. However, its effectiveness may be limited if this assumption does not hold, as adversarial clients constituting over half the clients could possibly sabotage the defense. The authors did not sufficiently discuss limitations to the mechanism or potential strategies if this threshold is exceeded.

2. The proposed method relies on applying the **Three Sigma Rule** to identify outlier clients, but does not adequately justify the underlying assumption that client behavior will follow a **Gaussian/normal distribution**. Without evidence or validation that the models meet this statistical requirement, the correctness and reliability of using this rule is unclear. The authors need to provide empirical or theoretical support for why these distributions were expected in this context, otherwise the key thresholding technique lacks rigorous foundation.

3. The paper claims to propose a new defensive mechanism, but its core algorithms - Krum, Three Sigma Rule and Zero-knowledge proofs - are already well-established in prior work. While integrating existing techniques can still yield new systems, the paper does not provide sufficient insight into how this combination offers meaningful advantage. Without novel algorithmic or analytical insights, the technical value of merely assembling known pieces is limited. To strengthen the paper, the authors should demonstrate deeper understanding of how this specific integration improves upon the state-of-the-art.

4. The evaluation of the proposed method's scalability and performance is limited by the small-scale datasets used (e.g. FEMNIST, CIFAR). While useful for proof-of-concept, these datasets do not adequately represent the data and computational heterogeneity of modern large-scale Federated Learning systems. To fully demonstrate the practicality and effectiveness of this defense, it will be important for the authors to test its performance and overhead when applied to real-world FL scenarios at a larger scale. Without such experimentation on industry-grade datasets and systems, the approach's scalability and real-world viability remain uncertain.

**Questions:**

Please refer to the Weaknesses section.

---

> ### Author Response · Authors · 2023-11-20
> **Response to Reviewer zU9y**
>
> We appreciated your constructed feedback that helped enhance our paper. Below are our responses.
>
> **W1: Assumption of Malicious Clients Remaining Below 50%**
>
> Thank you for highlighting the importance of discussing the limitations of our mechanism under different proportions of malicious clients. First, we would like to point out that security solutions often address scenarios that are less frequently encountered in the real world – those that may rarely occur, but when they do, they can lead to serious consequences. This holds true for security issues in Federated Learning (FL) systems as well. In practical FL systems, the proportion of malicious clients is typically no more than 10%. Consistent with this, most literature, to the best of our knowledge, evaluates scenarios with a maximum of 50% adversarial clients, as referenced in [1][2][3][4]. Some methods impose even stricter requirements on the percentage of malicious clients. For example, the well-known m-Krum approach [1] selects n out of N clients in each FL iteration, but only m out of these n local models contribute to the aggregation. The percentage of malicious clients must be less than 1/2 - (m+2)/n as required by the method, which is usually significantly lower than 1/2. For example, in the optimal scenario with 10 participating clients and only 1 local model contributing to the aggregation, which directly leads to an upper limit on the percentage of malicious adversaries when n is 10, the requirement is that the percentage of malicious clients is less than 1/2 - 3/10 = 20%.
> Your comment, however, has inspired us to explore further. If given the opportunity to enhance our paper, we plan to expand Experiment 5 (Varying the number of malicious clients) to include scenarios with over 50% malicious clients. This will allow us to test the limits of our approach in more extreme conditions. It is also noteworthy that our approach has performed well so far. In Experiment 5, where we varied the percentage of malicious clients to 0%, 20%, and 40%, the test accuracy remained very close to that of the benign scenarios, even with 20% or 40% malicious clients.
>
>
> [1] Blanchard, Peva, et al. "Machine learning with adversaries: Byzantine tolerant gradient descent." Advances in neural information processing systems 30 (2017).
>
> [2] Guerraoui, Rachid, and Sébastien Rouault. "The hidden vulnerability of distributed learning in byzantium." International Conference on Machine Learning. PMLR, 2018.
>
> [3] Karimireddy, Sai Praneeth, Lie He, and Martin Jaggi. "Byzantine-robust learning on heterogeneous datasets via bucketing." ICLR2022.
>
> [4] Xie, Chulin, et al. "Crfl: Certifiably robust federated learning against backdoor attacks." International Conference on Machine Learning. PMLR, 2021.
>
> **W2: Justification for Assuming Gaussian/Normal Distribution**
>
> We appreciate your feedback on the necessity of justifying the Gaussian distribution assumption for client models. As studied extensively in the distributed learning literature, if the local data of the clients are i.i.d., parameters of the local models follow a Gaussian distribution, see  [5][6][7]. Further, even if the client datasets are non-i.i.d., based on the Central Limit Theorem, the distribution of the local models tends toward a normal distribution.
>
> We included the following description in Section 3.2.
> ```
> The 3 Sigma Rule is pivotal for two reasons: i) in case the client datasets are i.i.d., parameters of local models follow a Gaussian distribution [5][6][7] ; and ii) even when client datasets are non-i.i.d., the Central Limit Theorem indicates that local models tend toward a normal distribution.
> ```
> [5] Baruch, Gilad, Moran Baruch, and Yoav Goldberg. "A little is enough: Circumventing defenses for distributed learning." Advances in Neural Information Processing Systems 32 (2019).
>
> [6] Chen, Yudong, Lili Su, and Jiaming Xu. "Distributed statistical machine learning in adversarial settings: Byzantine gradient descent." Proceedings of the ACM on Measurement and Analysis of Computing Systems 1.2 (2017): 1-25.
>
> [7] Yin, Dong, et al. "Byzantine-robust distributed learning: Towards optimal statistical rates." International Conference on Machine Learning. PMLR, 2018.

---

> > ### Author Response · Authors · 2023-11-20
> >
> > **W3: Novelty and Technical Value of the Proposed Defensive Mechanism**
> >
> > Thank you for pointing out the need for a clearer demonstration of the novel contributions of our work. Our solution handles anomaly detection in real-world FL systems, and existing state-of-the-art, e.g., Krum, does not work well due to the following reasons: 1) Krum keeps only one (or multiple for m-Krum) local model that is the most likely to be benign to represent the other local models in each FL iteration. This will completely waste the computation from the other benign clients and undermine the aggregation results. 2) Krum assumes attacks happen in each round of FL iteration. However, in real-world scenarios, attacks happen infrequently, and applying Krum will change the aggregation results even if there are no malicious clients. Considering these, we design a 2-step anomaly detection algorithm that first detects whether attacks have happened, and then, if attacks happened, we will further apply an anomaly detection procedure to remove the potential malicious client models. We utilize the 3 sigma rule to define a bound for the client models, so that the anomaly detection procedure removes a varying number of malicious client models based on the detection results, instead of removing a fixed number of client models. We also utilize zero-knowledge proofs to provide a verification to the clients so that  they ensure that the anomaly detection of the server is correct. With all these, our anomaly detection scheme offers novel contributions to the anomaly detection in FL systems.
> >
> > We included the following description in the introduction section:
> > ```
> > Moreover, existing defense mechanisms are deployed at the FL server without any verification procedures to ensure their correct execution. While most of the clients are benign and wish to collaboratively train machine learning models, they can also be skeptical about the server's reliability during the execution of the defense mechanisms that modify the original aggregation procedure.
> > ```
> > We also included the following description about Krum in the Related Work section:
> > ```
> > For each FL round, instead of using all client submissions for aggregation, such approaches keep local models that are the most likely to be benign to represent the other local models. Such approaches are effective, but they keep less local models than the real number of benign local models to ensure that all Byzantine local models are filtered out, causing misrepresentation of some benign local models in aggregation. This completely wastes the computation resources of the benign clients that are not being selected and thus, changes the aggregation results as some benign local models do not participate in aggregation.
> > ```
> >
> >
> > **W4: Evaluation of Scalability and Performance on Larger Datasets**
> >
> > Thank you for your feedback regarding the scalability and real-world applicability of our proposed method. We agree that testing on small-scale datasets may not fully capture the challenges of large-scale Federated Learning systems. To address this, we will extend our evaluations to include larger, more diverse datasets that are representative of real-world scenarios. We also included an experiment that evaluates our approach using real-world edge devices in the appendix. The description of the experiment is as follows:
> > ```
> > Exp8: Evaluations in real-world applications. We utilize edge devices from the Theta network (Theta Network., 2023) to validate the scalability of our anomaly detection approach to real-world applications. The FL client package is integrated into Theta’s edge nodes, which periodically fetch data from the Theta back-end. Subsequently, the FL training server capitalizes on these Theta edge nodes and their associated data to train, fine-tune, and deploy machine learning models. We utilize the Byzantine attack of random mode. Considering the challenges posed by real-world environments, such as devices equipped solely with CPUs (lacking GPUs), potential device connectivity issues, network latency, and limited storage on edge devices (for instance, some mobile devices might have less than 500MB of available storage), we choose a simple task by employing the MNIST dataset for a logistic regression task. We deploy 20 client edge devices, and set 5 as malicious for each FL training round. The information of the Theta devices is shown in Figure 15. The training process is shown in Figure 16, and the total training time is 221 seconds. We also include the CPU utilization and network traffic during training, which are shown in Figure 17 and Figure 18, respectively.
> > ```

---

> > > ### Comment · Reviewer_zU9y · 2023-11-23
> > > **Response to the rebuttal**
> > >
> > > Thanks for the authors’ response.

---

### Official Review · Reviewer_qLTB · 2023-10-30

**Soundness:** 3 good
**Presentation:** 3 good
**Contribution:** 3 good
**Rating:** 6
**Confidence:** 2

**Summary:**

The authors propose an anomaly detection approach for Federated Learning. Their method eliminates the need for prior knowledge of the number of malicious clients and avoids reliance on re-weighting or modifying submissions. Experimental results demonstrate the efficacy of their approach.

**Strengths:**

1. The paper is well written and generally easy to follow.
2. The method has been compared to a significant amount of related research.
3. A well-structured and clear presentation.

**Weaknesses:**

1.The explanations of some experimental results are not entirely convincing.
2.The description of the threat model needs to be more accurate.

**Questions:**

1.	It is preferable to describe the threat model based on adversary goals, knowledge, and capabilities.
2.	The justification provided for selecting the 'second-to-the-last layer' in Exp1 is not sufficient.
3.	I didn't fully grasp the significance of "VERIFIABLE ANOMALY DETECTION USING ZKP," and the authors should emphasize the research objectives of this section more prominently.
4.	The paper is very well structured and. There are occasional grammar hiccups and typos, so I recommend a light editing pass (below are a few of the mistakes I’ve collected, but there are more).
Page 5 In this ppaer –>in this paper

---

> ### Author Response · Authors · 2023-11-20
> **Response to reviewer qLTB**
>
> We would like to first express our gratitude for your reviews. Your insightful comments and constructive suggestions are invaluable in enhancing our work.
>
> Q1 & W2:The description of the threat model needs to be more accurate. It is preferable to describe the threat model based on adversary goals, knowledge, and capabilities.
>
> Thank you for your constructive insights. Based on your feedback, we modified the description of the threat model as follows:
>
> ```
> We consider an FL system where some clients are malicious, while most clients are honest. The clients have full access to their local data and can train models using their data. They also would like to collaboratively train a model without sharing their data. However, the malicious clients among them may conduct attacks to achieve some adversarial goals, including: i) planting a backdoor to misclassify a specific set of samples while minimally impacting the overall performance of the global model, i.e., backdoor attacks; ii) altering the local models to prevent the global model from converging, i.e., Byzantine attacks; and iii) randomly submitting contrived models without actual training, i.e., free riders. Clients are aware that the server can take some defensive methods to remove potential malicious local models, and they want to verify the mechanism is processed correctly and honestly at the server.
> ```
>
> Q2 & W1: The explanations of some experimental results are not entirely convincing. The justification provided for selecting the 'second-to-the-last layer' in Exp1 is not sufficient.
>
> Thank you for your suggestion. For the selection of the importance layer, our aim is to choose a general case that is suitable for real-world scenarios that involve various models and datasets. Intuitively, we selected the second-to-the-last layer,  as it contains more information and can represent the whole model. In Experiment 1, we observed that this layer exhibited higher sensitivity compared to most other layers across all scenarios, thus can be utilized to represent the whole model. We acknowledge that our initial presentation for Experiment 1 was unclear. To clarify this, we have revised our descriptions as follows.
> ```
> We utilize the norm of gradients to evaluate the "sensitivity" of each layer. A layer with a norm higher than most of the other layers indicates higher sensitivity compared with most of the other layers, thus can be utilized to represent the whole model. We evaluate the sensitivity of layers of CNN, RNN, and ResNet56. The results for RNN are shown in Figure 4, and the results for ResNet56 and CNN are deferred to Figure 14 and Figure 13  in Appendix B.2, respectively. The results show the sensitivity of the second-to-the-last layer is always higher than most of the other layers, which includes adequate information of the whole model, thus can be selected as the importance layer.
> ```
>
> Q3: I didn't fully grasp the significance of "VERIFIABLE ANOMALY DETECTION USING ZKP," and the authors should emphasize the research objectives of this section more prominently.
>
> We appreciate your feedback regarding the motivation of using ZKP. The intuition is that, while most of the clients are benign and wish to collaboratively train machine learning models, they may also have concerns about the server's procedure of removing malicious client models, as this procedure may modify the original aggregation result. Considering this, we deployed a ZKP-based verification procedure for the anomaly detection mechanism so that the benign clients can ensure the correct execution of the mechanism at the server.
>
> Based on your feedback, we included the following description at the beginning of Section 4 (Verifiable Anomaly Detection Using ZKP). We also included corresponding explanations in the introduction.
> ```
> This section introduces a ZKP-based verification procedure so that the benign clients can ensure the correct execution of the anomaly detection mechanism at the server. The intuition is that clients may be skeptical about the removal of some client models during the anomaly detection process at the server, as it may change the aggregation results.
> ```
> Q4: The paper is very well structured and. There are occasional grammar hiccups and typos, so I recommend a light editing pass (below are a few of the mistakes I’ve collected, but there are more). Page 5 In this ppaer –>in this paper
>
> Thanks for your feedback! We have proofread the paper multiple times and corrected the typos.

---

### Author Response · Authors · 2023-11-21

Dear Chairs and Reviewers,

We appreciate the time and effort you have invested in reviewing our paper. However, we would like to express our concerns regarding the review we received from Reviewer TEpF, which contains several inaccuracies that contradict the content of our paper. This reviewer gave an overall rating of 3 (reject) with a confidence level of 4 (confident in their assessment), and we find this review problematic for the following reasons: 1) The reviewer noted the absence of an important experiment, which, in fact, was indeed included in our original submission. 2) The reviewer recommended three specific papers in our experiments, however, one of these papers has already been used as a baseline in our paper and the remaining two are less feasible to be considered. 3) The summary appears to be a direct copy of our abstract, raising concerns about the thoroughness of the review.

We believe these issues merit attention as they may not align with the high standards of the ICLR reviewing process. We have provided detailed responses to each point raised by Reviewer TEpF for your consideration.

Thank you for your attention to this matter.

---

### Author Response · Authors · 2023-11-23

Dear Chairs and Reviewers,

We would like to express our concern regarding a toxic review we received from Reviewer TEpF that, unfortunately, appeared hasty and contained obvious mistakes controversial to our work. Though we submitted a detailed response to address these issues, Reviewer TEpF still did not acknowledge the shortcomings in their review. Instead, he/she advised us to "pay more attention to the paper itself," which did not address the basic issues we raised.

This leads to our great concern about Reviewer TEpF's expertise in reviewing papers for ICLR. Furthermore, we also would like to bring attention to the chairs and other reviewers on Reviewer TEpF’s possible negative impact on the ICLR community. As a member of this community, we believe that no one would like to have their months or even years of hard work to be directly rejected by a review that seems to have been hastily completed in just five minutes. Regardless of the acceptance or rejection of a single paper, the negative impact of such unprofessional behavior is prolonged and may potentially harm everyone in this community one day.

Regardless of the final decision of our paper, still we would like to thank Reviewer qLTB and Reviewer zU9y again for their valuable and professional reviews. We will integrate their constructive and professional feedback to improve our work in future revisions.

---

### Meta-Review · Area_Chair_HynP · 2023-12-01

**Metareview:**

This paper worked on federated learning and proposed a zero-knowledge-proof-based anomaly detection method to remove the malicious clients. There were two negative reviewers, pointing out many issues in its derivation (e.g., too strong assumptions) and experiments (e.g., missing closely related baselines). The authors didn't nicely address those issues. As a result, our reviewers consistently agreed to reject it (the only positive reviewer said that "I am willing to follow the decisions of the other reviewers").

Here is a forwarded message from the discussion (the original post was not visible to the authors but I think the message is quite helpful):
> In the review process, I raised some questions on the paper. I expected more comprehensive evaluations to make the paper more solid to meet the ICLR standards. However, the authors gave a lot of excuses instead of adding new analyses and results. Some of the excuses are obviously flawed. For example, the author said "To make the lines in the figure clear and easy to see, we decided to only include two settings and set the percentage of malicious clients to 20% and 40% only." Actually, they can include more results in the appendix. Instead of a single plot, multiple figures and tables will be more clear. This is not a convincing reason to not include more comprehensive evaluations. Moreover, DBA in ICLR2019 is a strong and stealthy backdoor baseline, which is widely used and cited. The authors just refused to compare with it.\
> Finally, I believe that discussions based on equal respect should be encouraged. The words of the authors to the reviewers do not show enough respect and may negatively influence the community. Anyway, the paper quality should always be the top 1 and I encourage the authors to pay more attention to the paper itself.

**Justification For Why Not Higher Score:**

There were two negative reviewers, pointing out many issues in its derivation (e.g., too strong assumptions) and experiments (e.g., missing closely related baselines). The authors didn't nicely address those issues.

**Justification For Why Not Lower Score:**

N/A

---

### Decision · Program_Chairs · 2024-01-16

Reject